# Overview of Atopic Dermatitis in Different Ethnic Groups

**DOI:** 10.3390/jcm12072701

**Published:** 2023-04-04

**Authors:** Andrea Chiricozzi, Martina Maurelli, Laura Calabrese, Ketty Peris, Giampiero Girolomoni

**Affiliations:** 1UOC di Dermatologia, Dipartimento di Scienze Mediche e Chirurgiche, Fondazione Policlinico Universitario A. Gemelli-IRCCS, 00168 Rome, Italy; 2Dermatologia, Dipartimento di Medicina e Chirurgia Traslazionale, Università Cattolica del Sacro Cuore, 00168 Rome, Italy; 3Section of Dermatology and Venereology, Department of Medicine, University of Verona, 37126 Verona, Italy

**Keywords:** atopic eczema, dermatitis, allergy, itch, skin disease, treatment, prevention, epidemiology, ethnic differences, cellular, molecular, immunological, physiological therapeutic

## Abstract

Atopic dermatitis (AD) is a common chronic inflammatory skin disease with a high prevalence worldwide, including countries from Asia, Africa, and Latin America, and in different ethnic groups. In recent years, more attention has been placed on the heterogeneity of AD associated with multiple factors, including a patient’s ethnic background, resulting in an increasing body of clinical, genetic, epidemiologic, and immune-phenotypic evidence that delineates differences in AD among racial groups. Filaggrin (FLG) mutations, the strongest genetic risk factor for the development of AD, are detected in up to 50% of European and 27% of Asian AD patients, but very rarely in Africans. Th2 hyperactivation is a common attribute of all ethnic groups, though the Asian endotype of AD is also characterized by an increased Th17-mediated signal, whereas African Americans show a strong Th2/Th22 signature and an absence of Th1/Th17 skewing. In addition, the ethnic heterogeneity of AD may hold important therapeutic implications as a patient’s genetic predisposition may affect treatment response and, thereby, a tailored strategy that better targets the dominant immunologic pathways in each ethnic subgroup may be envisaged. Nevertheless, white patients with AD represent the largest ethnicity enrolled and tested in clinical trials and the most treated in a real-world setting, limiting investigations about safety and efficacy across different ethnicities. The purpose of this review is to describe the heterogeneity in the pathophysiology of AD across ethnicities and its potential therapeutic implications.

## 1. Introduction

Atopic dermatitis (AD) is one of the most common immune-mediated skin disorders, with a chronic-relapsing course and a multifactorial pathogenesis [1]. The prevalence of AD is very high worldwide, including countries from Asia, Africa, and Latin America, across different ethnic groups, as well as across different ages [2].

The disease is clinically characterized by itchy eczematous lesions primarily involving flexural areas and the face, neck, and distal extremities, and it might precede other non-cutaneous atopic manifestations, such as asthma and allergic rhinitis (AR), priming the so-called “atopic march” [3].

Unlike other inflammatory skin diseases, AD shows a certain degree of clinical heterogeneity, leading researchers to consider this condition as a spectrum instead of a single entity. For this reason, in recent years, several efforts have been made to characterize disease subtypes according to different parameters, such as phenotypes, skin barrier status, IgE levels, age, gender, and ethnicity [4]. Moreover, different endotypes across different age groups and ethnicities and according to IgE levels and filaggrin mutation status have been identified through specific molecular signatures [5].

Depending on the patient’s racial background, AD seems to show different clinical, genetic, and immunopathogenic features [6]. Nevertheless, Caucasian patients with AD still represent the most studied ethnic group, leading to uncertainty in fully defining differences and similarities in disease pathophysiology across different ethnicities.

This review aimed to explore the current insights on AD heterogeneity depending on patient ethnicity and to analyze the possible therapeutic implications of these pathophysiologic differences.

## 2. Clinical Features

Several AD phenotypes have been described according to lesion distribution (e.g., flexural, head and neck, periorificial) or the predominant clinical feature, such as nummular eczema, prurigo nodularis, lichenified dermatitis, or follicular/papular dermatitis [7]. Interestingly, some differences in the clinical manifestations of AD exist among ethnic groups, which may pose diagnostic challenges. The clinical heterogeneity of AD is oftentimes the result of differences in the distribution and pigmentation of the lesions [8]. Asian individuals typically present lesions with more defined borders, sometimes closely resembling psoriasis plaques, as well as more scaling and lichenification, in comparison to white AD patients [9,10]. Conversely, African AD patients present with a predominant extensor involvement [11] and sometimes with perifollicular accentuation and distinct papules on the extensor surfaces and on the trunk [12,13]. Furthermore, in darker skin types exclusively, a lichen-planus-like presentation of AD has been described [14].

Because of the almost complete absence of erythema in dark skin types, oedema, warmth of the skin, or overlying scale may help dermatologists in identifying the presence of erythema. Furthermore, many scoring systems in clinical practice, such as the eczema area and severity index (EASI) [15] or the scoring AD (SCORAD) [16], rely on erythema to assess the disease’s severity and thus have the tendency to underestimate it in darker skin types [17]. In addition, patients with darker skin types, more frequently than white patients, exhibit xerosis, hyperlinearity of the palms, Dennie–Morgan lines, periorbital darkening, lichenification, and prurigo nodularis [11].

Finally, the risk of developing post-inflammatory hyper- or hypo-pigmentation is more common in dark skin [18].

## 3. Genetic Variability across Ethnic Groups

Similarly to other chronic inflammatory dermatoses, a genetic predisposition based on a polygenic background is associated with the development of AD. Certain genes have been implicated in the predisposition to AD, in particular those related to the skin barrier function, those involved in Th2 immune responses, and those implicated in the vitamin D metabolism and the synthesis of its receptors [19].

An attempt to study ethnic differences in AD from a genetic perspective, seeking to identify ethnic-specific genes, has been recently conducted in African, Asian, and Hispanic populations through genome-wide association studies (GWAS) [20,21,22,23,24,25,26].

Loss-of-function mutations (LOF) of the structural protein filaggrin represent the strongest genetic risk factor for the development of AD and are detected in up to 50% of European and 27% of Asian AD patients [27,28,29]. The filaggrin precursor pro-filaggrin is encoded by the *FLG* gene, located in the epidermal differentiation complex (EDC) on chromosome 1q23.3. Filaggrin deficiency is a major determinant of defective barrier function [30], and it is associated with (i) a disrupted keratinocyte differentiation, (ii) an impaired corneocyte integrity and cohesion, (iii) an altered tight junction formation, (iv) a decreased water retention, (v) an abnormal lipid formation, as well as (vi) an enhanced susceptibility to cutaneous infections [31]. Patients carrying LOF mutations in the *FLG* gene exhibit a more severe phenotype of AD, with a more persistent course, a higher risk of skin infections and allergic sensitization, as well as a major impairment of the immune system compared to patients with wild-type *FLG* [32,33].

Interestingly, filaggrin LOF mutations show a certain population specificity. The first two characterized mutations, R501X and 2282del4 [34], as well as S3247X and R2447X, have been extensively studied and are present in 7–10% of the white European population, whereas they are absent in Asian populations, which have a different spectrum of *FLG* variants [27]. Conversely, studies failed to detect common *FLG* LOF mutations in people of African ancestry with AD, although decreased levels of filaggrin have been reported in the skin of these patients [35,36]. Interestingly, no association between *FLG* LOF and AD have been found in African populations (Ethiopian) so far [37], whereas a relationship has been described in African Americans, probably due to genetic admixture [33,38]. Overall, *FLG* LOF mutations are six times less common in subjects of African descent than in people of European and Asian descent, suggesting a minor contribution to AD development [33].

The absence of *FLG* gene mutations in African people has prompted further genetic investigations in patients of this ethnicity. Results from whole-exome sequencing (WES) analysis in Ethiopian AD patients revealed an association between LOF mutations in the *FLG2* gene and the persistence of AD [39]. The *FLG2* gene is located within the EDC, sharing with *FLG* a similar molecular structure and biological functions [40,41]. Similarly, another study based on a whole exome analysis demonstrated that *FLG2* mutations were associated with a more persistent AD in an African American cohort [42]. On the contrary, studies conducted on European subjects did not reveal any association between the *FLG2* missense variant (rs16833974) and AD [43,44].

Beyond the EDC, variants in genes encoding for proteins involved in maintaining epidermal homeostasis have been described in specific ethnicities. For example, mutations in claudin 1 (*CLDN1*) gene, encoding for a structural protein of the tight junctions, have been related to the early onset of AD in Ethiopian patients, and it may be involved in the susceptibility of AD in this population [45]. In addition, some studies detected an association between LOF mutations in *SPINK5,* encoding for a serine-protease inhibitor involved in epidermal homeostasis and classically associated with both an autosomal recessive congenital ichthyosis and AD in East Asian populations [46,47,48,49].

The second class of genes implicated in AD susceptibility is related to the innate and adaptive immunity and, in particular, to the Th2 pathway that is central to AD pathogenesis.

Several studies described a correlation between polymorphisms in *IL-4, IL-13, IL-31, IL-4RA*, and *IL-31RA1* genes, all belonging to the Th2 pathway, and an increased risk of developing AD [50,51,52,53,54,55]. Similar to those involved in the epidermal function, genes related to the immune response characterizing AD showed some differences across ethnic groups.

In detail, polymorphisms of *IL-4* and *IL13/IL-13RA1* have been associated with an AD predisposition in Japanese, Korean, and Chinese populations [52,54,56,57], while AD susceptibility was significantly associated with *IL4RA* and *STAT6* polymorphisms in Egyptian children [54]. One specific polymorphism (Q576R) of the *IL4Rα* gene, associated with AD severity in murine models, has been found to be over-represented in the African American population [58,59].

Moreover, many other genes encoding molecules that contribute to AD pathogenesis at different levels, such as *STAT6*, thymic stromal lymphopoietin (*TSLP*), *IL7R, TSLPR*, interferon regulatory factor 2 (*IRF2*), toll-like receptor 2 (*TLR2*), FcεRIα (*FCER1A)*, and β-Defensin (*DEFB1*), have been found to been implicated in AD susceptibility in different ethnic groups [60,61,62]. For example, polymorphisms in the *IRF2* gene were associated with AD in European Americans (rs793814 and rs3756094) and in African Americans (rs3775572) [63], and a specific *TSLP* variant (rs1898671) has been related to less persistent AD in white and African Americans [64].

In Japan, a *TLR2* variant has been proposed as a genetic predictor of AD severity, while *FCER1A* polymorphisms have been related to significantly higher IgE serum levels in AD patients [65,66].

Variants in the *DEFB1* gene, encoding a small antimicrobial peptide mainly expressed in the epidermis and promoting host defense against pathogens, have been related to an increased risk of AD in Brazilian, Mexican, and Korean populations [67,68,69]. Notably, variants in genes encoding IL-12 and its receptor have been reported as susceptibility genes in Korean AD patients [70].

## 4. Pathogenesis

The pathogenesis of AD is characterized by a complex interplay between genetic and environmental factors contributing to epidermal barrier disruption, commensal skin microbiota dysbiosis, and alterations in immune responses, causing disease occurrence and/or exacerbation [3].

The contribution of the skin microbiota in AD is widely recognized [71]. In more than 90% of AD patients, skin lesions are colonized with Staphylococcus Aureus, and the density of the bacterium is associated with AD severity [72,73]. Similarly, AD skin during flares is characterized by a poor heterogeneity of the skin microbiota’s composition, favoring the predominance of S. Aureus [74]. Furthermore, S. Aureus skin colonization in infancy often precedes the diagnosis of AD [75], suggesting that it may be also a disease trigger. Differences in S. Aureus colonization and variability in the prevalence of specific bacterial virulence factors between ethnic groups have been reported and may be partially responsible for the molecular and phenotypic heterogeneity of AD across ethnicities, likely contributing to a more mixed immune dysregulation that involves not only type 2 inflammatory cells but also Th22 and Th17 cells [76]. Other environmental factors such as tobacco smoking habits, sun exposure, and psychosocial factors might negatively impact on disease onset or severity and they might differ across ethnicities. Racial and ethnic disparities have been detected in the use of tobacco [77]. Similarly, sun exposure differs based on ethnicity and cultural factors in subjects living in the same country [78]. Beyond chronic stressors that may impact any individual, psychosocial factors represented by racism, discrimination, and acculturative stress (anxiety or tension related to efforts to adapt to the values of dominant culture within a society) are uniquely frequent among ethnic minorities [79], with African American adults living below the poverty line being twice as likely to experience psychological distress and to not receive medical care [80].

### Immune Mechanisms Involved in AD Pathogenesis

The current pathogenic model of AD is centered on the activation of type 2 immune cells such as T helper (h)2 cells, T follicular helper (Tfh) cells, innate lymphoid cells 2 (ILC2), T cytotoxic (c)2 cells, eosinophils, mast cells, and basophils, with contributions from other pathways to different extents, including Th/Tc22, Th9, Th1, Th/Tc21, and Th17 [70,81,82].

During the initial phases of the pathogenic cascade, keratinocytes are activated by the excessive exposure to allergens, irritants, and microbial antigens due to an epidermal barrier disruption, causing the release of chemokines such as thymus and activation-regulated chemokine (TARC, known as CCL17) [81], macrophage-derived chemokines (MDC or CCL22) [82,83], and innate immune cytokines, such as IL-1β, IL-33, and thymic stromal lymphopoietin (TSLP), with subsequent activation of ILC2s and Th2 cell-mediated immune responses [84]. Particularly, TSLP-activated dendritic cells express OX40 ligand (OX40L) which, upon binding to its receptor OX40, induces Th2 differentiation of naive T cells, resulting in the production of Th2 cytokines (IL-4, IL-5, IL-13, and IL-31) [85]. Moreover, ILC2s represent a relevant source of IL-5 and IL-13, thus contributing to the creation of a type 2 inflammatory cytokine milieu [86].

While the type 2 signal dominates acute and chronic phases of the disease and is constantly elevated in AD in all ethnic groups, an upregulation of other immune pathways can occur differently across ethnicities [87,88].

The Asian endotype of AD is typically characterized by a strong Th17 signature, as well as by peculiar clinical and histological features [89]. Comparing the gene expression profile of European American, Japanese, and Korean AD patients, and including as a control groups patients with psoriasis and healthy controls, the presence of a strong Th2 activation was detected in both Asian and European American AD patients but not in psoriasis patients. Conversely, a significantly higher expression of Th17 and Th22 (IL17A, IL19, S100A12, IL-22) and lower expression of Th1 genes (CXCL9, CXCL10, IFNG) was found in Asian AD patients [10]. In addition, Asian AD skin showed greater acanthosis, higher Ki67 counts, and frequent parakeratosis, prompting the conclusion that the Asian AD endotype presents overlapping features with European AD and psoriasis [10].

Another study investigating the molecular profile of AD has been conducted on Chinese AD patients in comparison with European Americans, psoriatic patients, and ethnicity-matched healthy controls. The gene expression analysis showed similar results in Chinese AD patients compared with those observed in Japanese and Korean patients, suggesting a consistent Th17/Th2 or blended AD–psoriasis endotype across all Asian AD patients [90].

Along these lines, an upregulation of the serum Th2 markers was found in both Asian and European American AD patients, together with a lower expression of Th1 markers (IFNγ, CCL2, CCL3, CCL4) and increased Th22 activation in Asian patients compared to European American. In contrast to the gene expression signature detected in the skin, serum levels of Th17 markers were not increased in Asian patients [91]. Similarly, a characterization of the immunophenotype of AD in African Americans was recently carried out through gene expression analysis. The results from this study revealed a strong Th2/Th22-skewing in African American AD patients, with both Th2 and Th22 markers correlating significantly with disease severity, and concomitantly, an attenuation of the Th1 and Th17 axes compared to European Americans was observed. In addition, the skin of African American AD patients had peculiar barrier changes such as a lower filaggrin decrease but greater loricrin reduction that differed from European AD subjects [92].

## 5. Therapeutic Implications in Different Ethnic Subgroups

Currently, therapeutic approaches and recommendations for AD treatment do not differ across ethnic groups, nor is ethnicity taken into consideration in current European or American guidelines or recommendations [93,94,95]. Indeed, limited investigations on the effects of therapeutic agents on AD in different ethnic groups have been performed so far.

From daily clinical practice, it can be inferred that topical steroids are overall effective for all skin types. However, they should be used with caution as they frequently induce or worsen hypopigmentation in darker skin types [96].

A study on the use of pimecrolimus cream 1% on AD patients of different origins showed that ethnicity had no impact on treatment outcomes [97]. In addition, a pooled data analysis about the efficacy and safety data of tacrolimus ointment compared outcomes obtained in eight Asian countries with outcomes obtained in the United States, Europe, and Japan, reporting similar results [98].

For moderate-to-severe AD, phototherapy is a possible therapeutic option, especially narrow band UVB (NB-UVB). In one study conducted in Singapore, both NB-UVB- and UVA/NB-UVB-based phototherapy were effective in the treatment of AD in Asian children [99]. Of note, the use of phototherapy in darker skin types necessitates some special considerations: NB-UVB has been shown to require higher doses in darker skin [100,101], while UVA1 phototherapy seems to be equally effective in Fitzpatrick I-V skin phototypes without any dose adjustment [102].

Concerning systemic therapies, differences in terms of efficacy and safety profiles across ethnicities may depend on several factors, such as a different metabolism of the drug, which may ultimately affect its bioavailability. For example, black individuals have a 20–50% lower bioavailability of cyclosporine than white individuals, thus requiring higher doses of the drug [103]. The use of methotrexate, an antifolate metabolite acting as a immunosuppressive agent, has been associated with a higher risk of alopecia in black patients [103]. Furthermore, azathioprine, another immunosuppressive agent that requires the activity of the enzyme thiopurine methyltransferase (TPMT) for its metabolism, may cause severe toxicity in black patients at normal dosages, since deficiency of TPMT is prevalent in this population [103].

Dupilumab, a fully human monoclonal antibody blocking the shared IL4Rα subunit, binding to both IL-13 and IL-4, was the first biologic agent approved for the treatment of moderate-to-severe AD [104]. Phase III trials testing dupilumab included 20–27% Asian and 5–7% Black patients and suggested a comparable efficacy in diverse populations [105,106,107]. Recently, a post-hoc analysis from three phase III trials (LIBERTY AD SOLO 1, SOLO 2 and CHRONOS) assessed the efficacy and safety of dupilumab vs. a placebo across racial subgroups (White, Asian, Black/African American) and demonstrated that, independently of ethnicity, dupilumab resulted effective in improving AD and was deemed safe, with a favorable benefit–risk profile [108]. The identification of a specific polymorphism (Q576R) of the *IL4Rα* gene, overrepresented in the African American population, provides a further rationale for the great efficacy of dupilumab in this subgroup of AD patients [59].

Furthermore, a cross-sectional study assessing the patient burden and the impact on quality of life of AD in the U.S. population has shown that the effectiveness of dupilumab in improving the quality of life in patients was similar amongst various racial groups (white, Asian, black/African American) [96].

Novel biologic agents and small molecules, such as tralokinumab [109], upadacitinib [110], baricitinib [99], and abrocitinib [111], have been recently approved, and many others are currently under investigation for the treatment of moderate-to-severe AD.

Tralokinumab is a first-in-class IgG4 monoclonal antibody that specifically binds with high affinity to IL-13. Results from phase III clinical trials have demonstrated a substantial improvement in the severity and symptoms of AD after tralokinumab administration [112,113]. Recently, a sub-analysis of tralokinumab phase III trials (ECZTRA 1, 2, and 3) was conducted to evaluate its efficacy and safety in a North American population vs. a non-North American population. Of note, approximately 30–52% of patients in the North American population had skin of color, versus 5–25% in the non-North American population. The results of this post-hoc analysis of three large phase III trials displayed the safety and efficacy of tralokinumab regardless of the ethnicity [114].

## 6. Discussion

In recent years, a great interest in deciphering the clinical and molecular heterogeneity of AD across ethnicities has prompted the development of clinical, genetic, epidemiologic, and molecular studies analyzing AD in different ethnic groups, in an attempt to characterize the disease’s subtypes. Although the similarities among different ethnic groups are by far larger than the differences in all disease aspects, a growing body of evidence suggests race-specific alterations in the epidermal structure, as well as differences in the magnitude of upregulation related to certain immune pathways. In general, healthy skin from European, African, and Asian populations might exhibit significant molecular differences. For instance, the higher rate of *FLG* mutations in European populations could be attributed to evolutionary pressures as *FLG* deficiency may have provided enhanced immunity against infections, and therefore protection during European pandemics in past centuries [115], and it may have ensured greater vitamin D synthesis in the skin, conferring an evolutionary advantage at high latitudes [116].

Interestingly, even the atopic march, which encompasses food allergy, asthma, allergic rhinitis, and conjunctivitis after AD, may show a certain racial variability [117]. A longitudinal study indicated wide differences in allergic comorbid trajectories between black and white children with AD. Black children had higher risk of asthma and lower risk of allergic rhinitis and food allergy, while white children were most likely to progress to allergic rhinitis and food allergy, despite a lower asthma risk. These differences may be partly explained by ancestral genetic variabilities but also by different exposure to various race-specific environmental risk factors [118]. Overall, the presence of extracutaneous atopic manifestations suggest a prominent role of the type 2 inflammation in all ethnic groups. However, the contribution of other immune pathways led to the identification of endotypes corresponding to specific phenotypes that, so far, has been poorly defined, as demonstrated by the relatively high upregulation of the Th17 signal in Asian AD compared with European AD skin, which could explain the predominance of well-demarcated, psoriasiform lesions in Asian patients compared to European ones but cannot explain why the inhibition of IL-17A failed to demonstrate clinical, histopathological, and transcriptomic benefits (Figure 1) [106,119].

On the contrary, the efficacy of dupilumab first, and tralokinumab more recently, across ethnicities has confirmed the central role of type 2 inflammation in all AD ethnic groups. Notably, transcriptomic studies revealed broader effects of dupilumab in directly suppressing the type 2 inflammatory signal and indirectly down-regulating the Th17/Th22-related signals, partially explaining its efficacy in treating patients belonging to different ethnicities, even in those characterized by a lessened type 2 inflammatory signature [120]. African AD patients show strong Th2 skewing with a high correlation between disease severity and Th2/Th22-related markers as well as IgE serum levels [92]. Therefore, those patients may reasonably benefit the most from the neutralization of IL-22 (by fezakinumab, an anti-IL-22 monoclonal antibody) or a Th2-targeting agent.

Beyond selected cytokine antagonists, novel classes of drugs targeting Janus kinases (JAK) with different selectivity, thus having a “broader” range of action, demonstrated high efficacy in treating AD. The JAK inhibitors, targeting one or more members of the JAK family, can simultaneously inhibit several downstream pathogenic pathways in AD [121]. Their broad suppressive activity may be promising and appealing to provide therapeutic benefit across all ethnicities. No data are currently available on the safety and efficacy profile of JAK inhibitors across ethnicities because non-white patients are still under-represented in AD clinical trials (18% of all patients included are black, 6.9% Asian, and 2% Hispanic), estimating that only 59% of AD clinical trials performed between 2000 and 2009 included race and ethnicity as a baseline demographic characteristic, and this is a limitation for interpreting data through race [122,123,124].

## 7. Conclusions

Differences amongst various ethnicities in terms of clinical phenotypes and their corresponding immune endotypes exist, though they are not fully elucidated.

However, this ethnic heterogeneity of AD may hold important therapeutic implications.

The “one-size-fits-all” therapeutic approach offers results that are not always satisfactory and may not be ideal. Therefore, future investigations could hopefully stratify AD populations more precisely by ethnicity and pave the way for the development of a personalized and tailored approach that better targets the immunologic pathways involved in each ethnic subgroup.

Unfortunately, ethnicity data on most new agents for AD are largely not available, and the inclusion of different ethnic groups in randomized clinical trials, as well as sub-analyses by race, are of great importance and should be strongly encouraged.

## Figures and Tables

**Figure 1 jcm-12-02701-f001:**
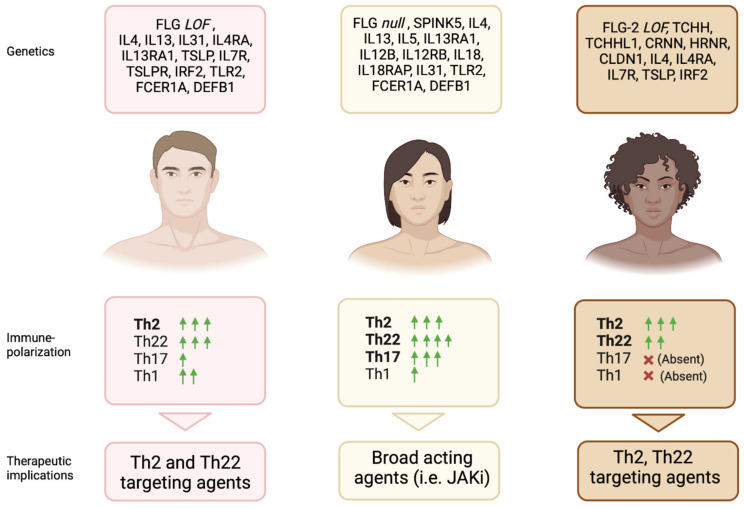
Main genetic, clinical, and immunological features in AD patients across ethnicities. Predominant genetic features of each subgroup, immune-polarization, and potential therapeutic implications are shown. FLG: Filaggrin; LOF: Loss-of-function; IL: Interleukin; TSLP: Thymic stromal lymphopoietin; IRF2: Interferon regulatory factor 2; TLR2: Toll-like receptor 2; FCER1A: Fcε Receptor1α; DEFB1: B -Defensin 1; SPINK5: Serine Peptidase Inhibitor Kazal Type 5; TCHH: Trichohyalin; TCHHL1: Trichohyalin like 1; CRNN: Cornulin; HRNR: Hornerin; CLDN1: Claudin 1; Th: T helper; JAKi: JAK inhibitors. Created with BioRender.com.

## Data Availability

Not applicable.

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
