# Peer review of "Overview of Atopic Dermatitis in Different Ethnic Groups"

_jcm, 2023, doi:10.3390/jcm12072701_

Round 1

Reviewer 1 Report

The title should be changed to "overview" rather than "pathophysiology", however, I leave the final decision here to the authors.

The information in the abstract part of the article is sufficient and satisfactory.

Especially the explanation of the title “Genetic variability across ethnic groups” was detailed and informative.

The Treatment title can be expanded a little more. Discussion and conclusion sections are okay.

There are some minor spelling errors in the article. It should be corrected.

Author Response

Review Report (Reviewer 1)
The title should be changed to "overview" rather than "pathophysiology", however, I leave the final decision here to the authors.
The information in the abstract part of the article is sufficient and satisfactory.
Especially the explanation of the title “Genetic variability across ethnic groups” was detailed and informative.
The Treatment title can be expanded a little more. Discussion and conclusion sections are okay.
There are some minor spelling errors in the article. It should be corrected.

Answer: 
We thank very much Reviewer 1 for the comments on the manuscript and the overall positive feedback. We added further details in the subparagraph previously entitled “Therapeutic implications”, though there is limited evidence about the impact of ethnicity on the therapeutic response or treatment approach in AD. Please find our changes on page 6 of 17 lines 277-279 (highlighted in yellow).
We also changed the title accordingly. 
Further, we edited the text for spelling errors. 

Reviewer 2 Report

In the current manuscript entitled “Pathophysiology of atopic dermatitis in different ethnic groups” the authors discuss about the heterogeneity of AD pathophysiology across different ethnicities. Even though, the authors pointed out that resemblances are far larger than difference in AD patients but raise a valid point about the limitation, safety, and efficacy of current treatment plans for AD patients among different ethnic groups.

Major concerns:

1.     Besides genetic variations, external environmental factors also contribute to the AD pathogenesis. For instance, S. aureus skin colonization is common in AD, which aggravates the disease and may contribute to its pathogenesis. Furthermore, S. aureus skin colonization in infancy precedes the diagnosis of AD, suggesting that it may be also a disease trigger (PMID: 28842320). Thus, the authors need to discuss the potential role of S. aureus skin colonization among different ethnic groups which might skew the Th2 to Th2/Th17 dominated AD. 

2.     The authors discussed about some genetic variant is associated with AD severity (line: 147-54). However, the authors further need to extend their discussion among other reported variant and disease severity. For instance, a recent report suggests that the IL-4Rα Q576R polymorphism, which is common in the African American, is associated with disease severity (PMID: 36690254).

Minor concern:

The quality of figure needs to change.

Author Response

Review Report (Reviewer 2)

In the current manuscript entitled “Pathophysiology of atopic dermatitis in different ethnic groups” the authors discuss about the heterogeneity of AD pathophysiology across different ethnicities. Even though, the authors pointed out that resemblances are far larger than difference in AD patients but raise a valid point about the limitation, safety, and efficacy of current treatment plans for AD patients among different ethnic groups.

Major concerns:
1.     Besides genetic variations, external environmental factors also contribute to the AD pathogenesis. For instance, S. aureus skin colonization is common in AD, which aggravates the disease and may contribute to its pathogenesis. Furthermore, S. aureus skin colonization in infancy precedes the diagnosis of AD, suggesting that it may be also a disease trigger (PMID: 28842320). Thus, the authors need to discuss the potential role of S. aureus skin colonization among different ethnic groups which might skew the Th2 to Th2/Th17 dominated AD. 

Answer: We thank Reviewer 2 for the insightful comment and for highlighting this important pathogenic aspect. We added on page 4 of 17, lines 168-187, highlighted in yellow, a focus on some environmental factors involved in AD that can differ across ethnicities. Because of the length and the distinct topic discussed below the description of environmental factors, we also subdivided that paragraph with a subheading entitiled: “4.1 Immune mechanisms involved in AD pathogenesis”

 2.     The authors discussed about some genetic variant is associated with AD severity (line: 147-54). However, the authors further need to extend their discussion among other reported variant and disease severity. For instance, a recent report suggests that the IL-4Rα Q576R polymorphism, which is common in the African American, is associated with disease severity (PMID: 36690254).
Answer: We thank Reviewer 2 for this relevant comment. We included additional information on the IL-4Rα Q576R polymorphism which has been associated with AD severity in murine models (page 6 of 17, lines 277-279).

Minor concern:
The quality of figure needs to change.

Answer: We have uploaded a new high-quality figure.

Round 2

Reviewer 2 Report

Thank you for addressing the comments.